# Determinants of Simultaneous Use of Soil Fertility Information Sources among Smallholder Farmers in the Central Highlands of Kenya

**Pamellah A. Asule** [1,*]**, Collins Musafiri** [2] **, George Nyabuga** [3]**, Wambui Kiai** [3]**, Felix K. Ngetich** [2,4]
**and Christoph Spurk** [5]

[1] Department of Water and Agricultural Resource Management, University of Embu,
    Embu P.O. Box 6-60100, Kenya
[2] Cortile Scientific Limited, Nairobi P.O. Box 34991-00100, Kenya; collins.musafiri15@gmail.com (C.M.);
    felixngetich@gmail.com (F.K.N.)
[3] Department of Journalism and Mass Communication, University of Nairobi,
    Nairobi P.O. Box 30197-00100, Kenya; george.nyabuga@gmail.com (G.N.); wamkiai@gmail.com (W.K.)
[4] School of Agricultural and Food Sciences, Jaramogi Oginga Odinga University of Science and
    Technology (JOOUST), Bondo P.O. Box 210-40601, Kenya
[5] Institute of Applied Media Studies, Zurich University of Applied Sciences, 8400 Winterthur, Switzerland;
    spurk@protonmail.com
[*] Correspondence: asule.pamellah@gmail.com; Tel.: +254-720381056

**Abstract:** Soil fertility decline is a significant drawback to food and nutritional security in sub-Saharan Africa. However, information and knowledge barriers seriously impede the adoption, effective use, and scaling up of soil fertility management innovations, especially by smallholder farmers who produce the bulk of the region's food needs. Apart from the knowledge that smallholder farmers seek soil fertility information from diverse sources, which they apply simultaneously, there is limited knowledge of farmers' information-seeking behaviour regarding which sources are used simultaneously and the factors influencing these choices. We employed a cross-sectional survey study design to determine the simultaneous use of soil fertility information sources of 400 smallholder farming households in the Central Highlands of Kenya. We analysed the data using descriptive statistics, principal component analysis (PCA), and a multivariate probit model. The PCA distinguished seven categories of information sources farmers use: local interpersonal, cosmopolite interpersonal, aggregative, print/demonstration, broadcast media, community-based, and progressive learning sources. The intensity of use revealed that most of the smallholders used soil fertility information sources simultaneously and primarily as complements. The determinants of simultaneous use of soil fertility information sources were farmer location, marital status, main occupation, age, farming experience, exposure to agricultural training, group membership, arable land and livestock units owned, soil fertility status, soil fertility change, and soil testing. This study's findings have implications for information dissemination strategies involving using multiple complementary sources of knowledge for improved soil health and productivity.

**Keywords:** soil fertility information sources; soil fertility decline; dissemination; simultaneous use; heptavariate probit model



## 1. Introduction

Soil plays an important role in sustaining the world's agroecosystems. Approximately 98 per cent of the human population depends on food derived from the soil [1]. Declining soil fertility, however, is a major threat to global agricultural food systems [2–4]. Poor soils limit the capacity of agroecosystems to meet the demand for food by the world's population of 7.9 billion, which is projected to rise to about 10 billion by 2050 [5]. In sub-Saharan Africa (SSA), the main cause of soil fertility decline is continuous cropping with

minimal nutrient replenishment [6]. The principal soil fertility constraints in the region include deficiencies in nitrogen and phosphorus, acidity, and low organic carbon content [7]. However, the delivery of recommendations to farmers to address these problems is a major impediment to soil fertility improvement strategies. Consequently, knowledge barriers constitute an important limitation in the adoption, effective use, and scaling up of soil fertility management innovations in SSA [8,9].

Several studies have reported the successful development, testing, and validation of soil fertility management practices with proven potential to enhance soil health and productivity in developing countries, including Kenya [10–14]. Related studies also indicate that many of the soil fertility ameliorating practices recommended to farmers are economically viable in diverse farming systems; such as the use of quality organic inputs and their integration with inorganic fertilisers [15–17]. Despite the evidence of the availability of technologies whose implementation can enhance productivity and the food and nutritional security of smallholder farmers, the level of adoption is below par in the central highlands of Kenya [18–20].

Smallholder farmers in SSA, including the central highlands of Kenya, have diverse needs for information, including the technical knowledge required to implement recommended soil fertility management technologies and practices, such as integrated soil fertility management or ISFM [21,22]. Sustained exposure to sources of learning by farmers seems to enhance the adoption of soil fertility technologies [23]. However, farmers in different regions face various challenges when it comes to the acquisition of agricultural information. These include limited access to sources of information, low awareness of relevant sources, language barriers, limited technical knowledge of sources, uncoordinated delivery of information, illiteracy, and financial constraints, among others [8,24–27]. Farmers' capacity to access the required agricultural information is further undermined by a lack of skills to utilise available sources of information effectively, especially under the pluralistic and demand-led extension frameworks promoted by countries in SSA [28–31]. Given the knowledge-intensive nature of technologies for soil fertility management, studies aimed at understanding how farmers use information sources, including factors influencing their information-seeking behaviour, are necessary to inform appropriate interventions.

Farmers use diverse sources of agricultural information and knowledge simultaneously to harness the complementary benefits of different sources [32,33]. Public agricultural extension agents are crucial for introducing complex ISFM recommendations and training farmers on the implementation of new technologies [30,34]. Extension programmes can be enhanced by using methods that encourage more farmer involvement in the learning process, promoting peer-to-peer learning, and offering opportunities for farmers to interact with other sources of knowledge including researchers, lead farmers, agro-dealers, and local leaders, among others [30,35]. Some of the methods and approaches being adopted to enhance extension activities include learning centres, farmer-to-farmer extension, demonstrations, and farmer field schools, among others. Farmer-to-farmer extension has high sustainability potential in addition to its effectiveness in encouraging the uptake of new technologies, especially where there is technical backstopping of the farmer trainers by extension workers [36,37]. Farmers continuously draw upon their knowledge and experience to address specific farm-level soil fertility constraints [38,39]. However, there is concern that farmers' indigenous and experiential knowledge is hardly adequate for addressing new agricultural challenges, which require modern technologies and best agronomic practices [40]. Researchers and practitioners alike, therefore, recommend farmer-participatory research approaches because they combine local knowledge and farmer experimentation, scientific expertise, and new knowledge to generate technologies with both technical feasibility and adaptation to farmers' conditions [41].

Modern ICT-enabled sources which utilise online platforms can be used to introduce new knowledge and technologies, and hence, cover for the shortage of extension personnel, but face-to-face interactions are still essential for confidence building when it comes to complex technologies like those required for soil fertility management [41].

Once considered a one-way channel, radio has become a two-way channel suitable for giving instant feedback to farmers due to the integration of mobile SMS and phone-ins in agricultural radio programmes [42]. The effectiveness of this medium is further enhanced in programmes where extension workers and other experts, including experienced farmers, are invited to offer accurate technical knowledge and experiential advice not possessed by radio station staff [43]. Printed agricultural information materials, when used alongside other information sources, offer the required reference material for farmers to learn from at their own pace, especially when implementing difficult agricultural practices [44]. The successful application of printed media sources of agricultural information depends on factors like quality of information, newness, timely delivery, and farmer interest [45].

Non-governmental organisations (NGOs) are better resourced than public extension sources and hence, capable of applying more participatory information dissemination approaches [31]. However, factors such as narrow mandates that may be different from farmers' needs, low sustainability, and poor reach limit the effectiveness of NGOs as agricultural information sources for farmers [29,31]. Partnerships between community development organisations and national agricultural extension institutions have also shown a positive influence on farmers' uptake of sustainable agriculture technologies due to enhanced access to services and improved capacity [46]. Overall, the utility of channels for ISFM information seems to depend on farmers' changing needs across the agricultural product value chain with frequency of channel use and usefulness of the information shared being key considerations as well [30].

Studies of the simultaneous use of agricultural information sources highlight demographic, socio-economic, institutional, and locational influences on farmers' choices. Specifically, the factors which seem to influence farmers' information-seeking behaviour include the age of the farmer, education, household size, market access, access to information assets, access to credit, electricity, and knowledge of formal sources of information, among others [32,47]. These studies considered sources such as ICTs, traditional media, field extension, social networks, face-to-face sources, and other farmers. Whereas an understanding of farmers' information-seeking behaviour and influencing factors is essential in designing farmer-oriented information and knowledge policies, there is a paucity of relevant data on smallholder farmers in SSA and specifically the Central Highlands of Kenya. This kind of information is necessary for enhancing the use of scarce information resources by targeting specific farmer groups using a combination of sources that are likely to be effective in reaching them. We conducted this study to investigate the simultaneous use of sources of soil fertility information, determinants, and barriers.

This study will contribute to the literature on the factors affecting the simultaneous use of information sources in SSA concerning a knowledge-intensive practice such as soil fertility management. The results of this study could have implications on the demand-led and pluralistic extension policies and practices by signalling where to direct farmers' information needs and relevant feedback to reach the right farmer demographics more effectively. Understanding farmers' information sources will make it possible to tailor the dissemination of information to the needs of different types of farmers. Additionally, smallholders will gain access to soil fertility information that could enhance soil productivity and hence, guarantee food and nutritional security.

## 2. Materials and Methods

### 2.1. Description of Study Sites

In this study, we collected data on the soil fertility information-seeking behaviour of smallholder farmers from Gatanga and Meru South sub-counties located in Murang'a and Tharaka-Nithi counties, respectively. The two counties are situated within the Central Highlands of Kenya. The sub-counties share most agroecological zones (AEZs), cropping activities, and land use practices [3]. They experience bimodal rains with the long rains season coming between March and June and the short rains from October to December. Gatanga sub-county lies within five AEZs: Lower Highlands (LH1), Upper Highlands

(UH1), and Upper Midlands (UM1, UM2, and UM3) as described in Jaetzold et al. [48]. The sub-county receives annual rainfall ranging between 900 and 1400 mm. Meru South sub-county lies within eight AEZs, namely Lower Highlands (LH1), Upper Midlands (UM1, UM2, and UM3), Lower Midlands (LM3, LM4, and LM5), and Intermediate Lowlands (L5) as described in Jaetzold et al. [48]. The long-term annual rainfall received in this region ranges between 600 and 1800 mm with a daily average temperature of 20 °C [48].

Farmers in the Central Highlands of Kenya cultivate both food and cash crops integrated with livestock keeping on small parcels of land [18,49]. The cash crops grown in the area include coffee, tea, tobacco, napier grass, and banana, while maize is the primary food crop. Farmers also cultivate beans, peas, sorghum, and millet. Gatanga sub-county has a population of about 187,987 persons, 55,461 households, and a population density of 354 people per square kilometre [50]. Meru South has a population of 144,290 persons, 42,594 households, and a population density of 312 persons per square kilometre. The high population density in the two regions has resulted in land fragmentation with the practice of continuous cropping exposing the need for measures aimed at soil fertility improvement to sustain production.

### 2.2. Sampling and Data Collection

The target population of our study consisted of smallholder farmers in the Central Highlands of Kenya. We employed a cross-sectional survey research design to collect data from household heads identified through a multistage sampling procedure. First, we conducted a literature review of soil fertility technologies development, adoption, and information needs to select the counties and sub-counties of interest. Second, we performed a whole sampling to select all the wards at the sub-county level. Third, we used a proportionate-to-size sampling procedure to determine the number of households to include in our sample from each ward. Fourth, we randomly sampled the individual household heads for inclusion in the study using a sampling frame obtained from agricultural officers at the ward level.

We determined the sample size for each sub-county following the formula shown in Equation (1) as described by Cochran [51].

$$n = \frac{z^2 pq}{E^2} = \frac{1.96^2 \times 0.5(1 - 0.5)}{0.0693^2} = 200 \tag{1}$$

where: n = sample size, z = z-value (e.g., 1.96 for 95% confidence level), p = percentage picking a choice, expressed as a decimal (0.5), q = 1 − p, and E = 6.93% allowable error expressed as a decimal (0.0693). Therefore, we sampled 200 households from each sub-county, resulting in a sample size of 400 households.

We recruited ten enumerators per site to assist in data collection based on fluency in the local dialects of their respective regions and familiarity with the cultural norms and practices relating to interaction with strangers and older persons. In addition, we selected enumerators who could communicate fluently in English and subjected them to training on techniques of data collection. Before the actual data collection exercise, we pre-tested the research instrument with 15 randomly selected respondents from each study site. The pre-test targeted the quality of the questionnaire, the enumerators' ability to administer it in an actual field situation, and the efficacy of the data-capturing technology. Each enumerator had a tablet equipped with Open Data Kit (ODK) software Version 1.4.9 for capturing the required data from the respondents. The questionnaire contained questions on the demographic characteristics of farmers, household socio-economic factors, farmers' assessment of the fertility status of their agricultural land, soil fertility information needs, and the sources used to obtain information on soil fertility management, among other issues. The enumerators administered the questionnaire to the male or female head of the sampled households using the face-to-face interview approach.

*2.3. Dependent Variables*

In our survey, we asked all the respondents to enumerate their sources of information and knowledge on soil fertility management. The respondents were also required to state the frequency of use of each mentioned source. We used the dummy variable 1 to denote that a given farmer used a specified source of soil fertility information and 0 otherwise. The farmers' responses yielded 25 sources.

In our subsequent analysis, we subjected the 25 sources of soil fertility information mentioned by farmers to principal component analysis (PCA) in SPSS 24 software to reduce them to a few non-correlated principal components (PCs). Before PCA, we checked the data for outliers using boxplots. We also checked for missing values and reduced the entries from 400 to 397. The principal component analysis (PCA) revealed a Kaiser-Meyer-Olkin (KMO) measure of sampling adequacy of 0.74 and Bartlett's Test of Sphericity statistic of $p = 0.0000$. The KMO was greater than 0.5, and Bartlett's Test of Sphericity was statistically significant, thus signifying that the dimensional data reduction was credible [52,53]. We performed an orthogonal rotation (Varimax method) to extract the loadings matrix. We used loadings greater than 0.5 for the interpretation of the PCs [54].

Our data revealed seven distinct categories (PCs) of information sources used by farmers to learn about soil fertility management. The seven PCs, which constituted the dependent variables of our study, were as follows: local interpersonal sources, cosmopolite interpersonal sources, broadcast media sources, aggregative/modern ICT-based sources, print media/demonstration sources, community-based sources, and progressive learning sources. The dependent variable was denoted as 1 if a farmer obtained soil fertility information from at least one source in the specified PC or 0 otherwise (Table 1).

**Table 1.** Dependent and predictor variables hypothesised to influence smallholder farmers' information-seeking behaviour.

| Variable Description | Code | Unit | Expected Sign $\pm$ |
|---|---|---|---|
| **Dependent variables** | | | |
| Local interpersonal sources (1 Yes, 0 No) | PC 1 | % HHs [a] | |
| Cosmopolite interpersonal sources (1 Yes, 0 No) | PC 2 | % HHs | |
| Modern ICT-based sources (1 Yes, 0 No) | PC 3 | % HHs | |
| Print/demonstration sources (1 Yes, 0 No) | PC 4 | % HHs | |
| Broadcast media (1 Yes, 0 No) | PC 5 | % HHs | |
| Community-based sources (1 Yes, 0 No) | PC 6 | % HHs | |
| Progressive learning sources (1 Yes, 0 No) | PC 7 | % HHs | |
| **Location** | | | |
| Household from Murang'a or Tharaka-Nithi county (1 if Tharaka-Nithi; 0 if Murang'a) | Site | % HHs | $\pm$ |
| **Predictor: Characteristics of household and household head (HHH)** | | | |
| Gender of household head (HHH); 1 male, 0 female) | HHH male | % HHs | + |
| Education level of HHH (0 no formal education, 1 primary and above) | HHH literate | % HHs | + |
| Marital status of HHH (1 married, 0 otherwise) | HHH married | % HHs | + |
| Main occupation of HHH (1 agriculture, 0 otherwise) | HHH agriculture main occupation | % HHs | + |
| Age of HHH (years) | HHH age | years | - |
| Household size | HH size | number | - |
| Farming experience of HHH (years) | HHH farming experience | years | + |

**Table 1.** *Cont.*

| Variable Description | Code | Unit | Expected Sign $\pm$ |
|---|---|---|---|
| **Predictor: Socio-capital attributes** | | | |
| The land had a title deed (1 Yes, 0 No) | Land secured | % HHs | + |
| HHH accessed agricultural training (1 Yes, 0 No) | Agricultural training | % HHs | + |
| HHH was a member of the agricultural group (1 Yes, 0 No) | Group membership | % HHs | + |
| **Predictor: Household resources** | | | |
| Land size under cultivation (acres) | Arable land size | acres | + |
| Livestock owned by household (Tropical livestock units) | Tropical livestock unit (TLU) | TLU [b] unit | + |
| **Predictor: Perception of soil fertility** | | | |
| Soil fertility poor (1 Yes, 0 No) | Soil fertility poor [c] | % HHs | - |
| Soil fertility moderate (1 Yes, 0 No) | Soil fertility moderate | % HHs | $\pm$ |
| Soil fertility good (1 Yes, 0 No) | Soil fertility good | % HHs | + |
| Soil fertility declining (1 Yes, 0 No) | Soil fertility declining [d] | % HHs | - |
| Soil fertility stable (1 Yes, 0 No) | Soil fertility stable | % HHs | $\pm$ |
| Soil fertility improving (1 Yes, 0 No) | Soil fertility improving | % HHs | - |
| Farmer's soil was tested (1 Yes, 0 No) | Soil tested | % HHs | + |

[a] Percentage of households; [b] tropical livestock units equivalent of cattle (0.7), sheep (0.1), goat (0.1), pig (0.2), and chicken (0.01) [53,55]. [c] Households with soil fertility status poor are the reference point. [d] Households with soil fertility declining are the reference point.

### 2.4. Explanatory Variables

We based the selection of our explanatory variables on a review of relevant studies of farmers' information-seeking behaviour concerning their agricultural activities and the implementation of new technologies and practices. Several studies support the hypothesis that farmer demographics, household socio-economic factors, perceptions of soil quality, and location are key determinants of the decision to use (or not use) certain information sources [22,32,33]. Smallholders decide whether to use soil fertility information based on the expected utility. The utility maximisation theory. If the utility of seeking soil fertility information is greater than the traditional, the smallholders will seek information using that channel. Table 1 shows the predictor variables used in our study and the expected impact on the selection of sources of soil fertility information by farmers in the Central Highlands of Kenya. Our selection of the independent variables was informed by the literature evidence regarding their influence on information utilisation and implementer of innovative practices. For instance, Foguesatto et al. [54] highlighted 80 variables that could significantly influence smallholders' decision-making process based on expected utility. Baumgart-Getz et al. [55] categorised the key independent variables as capacity, attitude, and environmental. For brevity, we selected the variables included in the questionnaire-based literature on the expected sign (+/−).

We considered seven variables to determine the influence of farmer and household characteristics on the simultaneous use of soil fertility information sources. Age was predicted to have a negative effect on the use of information sources based on empirical evidence from studies such as [32,56]. Older farmers tend to shun agricultural improvement measures in favor of farming activities with more immediate benefits, Yaseen et al. [57], perhaps because of their short planning horizon [46]. Age was an essential factor in our

analysis because of the study's relatively high mean age of farmers. We predicted a positive relationship between farmer education and information sources because educated farmers were more likely to search for and process information [46,57]. Household size promotes the simultaneous use of information sources because of the possibility of pooling together household members' networks, information resources, and skills. We included a gender variable in the analysis because of reported gender differentials in access to agricultural information where women farmers are constrained mainly by social, cultural, and economic barriers [58]. Identifying possible information sources for women farmers was crucial, and the study predicted negative and positive outcomes. Married household heads were likely to use information sources simultaneously because of joint decision-making for better livelihood outcomes. We predicted that agriculture as a primary occupation enhanced the simultaneous use of soil fertility information sources because farmers sought to maximise productivity. Experienced farmers tend to be stable with their time-tested practices and may not see the need for learning new practices; hence, the predicted sign for farmer experience was negative.

The socio-capital factors in our equation were land tenure security, access to agricultural training, and group membership. Ownership was vital for farmers to invest in long-term improvements on their land, including soil fertility management. Thus, farmers with title deeds were likely to be innovative and to seek information from various sources. Group membership increases opportunities for farmers to access diverse information resources, including agricultural extension, peer-to-peer learning, community radio, and ICTs, in convenient and cost-effective methods [52].

The two factors used as proxies for farmers' economic status, namely the size of arable land and livestock ownership expressed as the number of TLUs, were predicted to be positively associated with selecting information sources.

Farmers use their perceptions of soil fertility conditions to make management decisions [33]. Those with perceived fertile soils may use information sources to maintain this status [21]. Farmers with perceived poor soils may also want to apply technologies for improving such soils and could increase their use of information sources. Farmers with perceptions of infertile soils paid little attention to the problem [33]. Likewise, those who perceive declining soil fertility conditions tend to seek information on methods of reversing this trend. Given the various scenarios, this study predicted mixed results for using information sources in response to farmer perceptions of soil fertility conditions. On the other hand, soil testing may stimulate the search for additional information to implement the recommendations, hence, the prediction of a positive association with information seeking. Land size and the number of livestock units owned on farmer resources were hypothesised to enhance the use of information sources.

### 2.5. Empirical Modelling: Multivariate Probit Model

Farmers in our study could access diverse sources of information and knowledge for their soil fertility management decision-making. We predicted that a farmer needing soil fertility information could use one or more of the available sources. Hence, univariate or multivariate models were applicable for deciphering patterns of usage of information sources. The univariate model assumes that the decision to seek information from a given source does not depend on the other sources at the farmer's disposal, including those already in use. Thus, the univariate analysis approach treats soil fertility information-seeking behaviour, in terms of source selection, as a dichotomous choice to use (or not use) a specific information source. This approach does not consider the possibility of simultaneous use of multiple sources of information.

On the other hand, the multivariate models assume that the decision to seek agricultural information from a given source could be enhanced or discouraged by the choice to use other sources [32,47,59]. Further, the relationship between farmers' information sources could be either complementary (positive correlations) or substitutive (negative correlations), as demonstrated by [59]. This study assessed the likelihood of an explanatory

variable being a determinant of information seeking as opposed to the odds of success of information seeking as a result of independent variables. Therefore, the probit model was more justified than the logit model. Accordingly, our study employed a multivariate model (heptavariate probit, HVP) to assess the possibility of simultaneous use of information sources by farmers in our study area. This model allowed for the free correlation of unobserved and unmeasured error terms.

We performed the HPV modelling in Stata 15 software. This study structured the HPV equations for synchronous information-seeking behaviour [57]. We hypothesised that a farmer could seek information from a given source when the utility of using the source was greater than not using it. Therefore, the utility of seeking information from a given source is a latent variable determined by observed characteristics and the heptavariate distribution of the error terms described in Equation (2).

$$U_{ik}^* = X_i B_k + \varepsilon_i (k = LI, CI, A, PV, BM, CB, PL)　　　　　　(2)$$

where $U_{ik}^*$ is the net utility of seeking soil fertility information from the kth source; $X_i$ is a vector of observed household characteristics; $B_k$ is a vector of the coefficients to be estimated; LI, CI, A, PV, BM, CB, and PL refer to local interpersonal, cosmopolite interpersonal, aggregative/modern ICT, print/demonstration, broadcast media, community-based, and progressive learning information sources, respectively; and $\varepsilon_i$ refers to normally distributed heptavariate error terms. According to the utility maximisation theory, farmers will only seek soil fertility information from a given source if the gains from using that source outweigh not seeking information from it. Therefore, this is a binary choice, as described in Equation (3). We used Pearson's correlation to assess the relationship between dependent and independent variables since our data was not ranked. Spearman's correlation is implemented when the variables are ordinal or ranked.

$$U_{ik}^* = \begin{cases} 1 \text{ if } U_{ik}^* > 0 \\ 0 \text{ otherwise} \end{cases}　　　　　　(3)$$

In this study, we tested the null hypothesis that the pairwise HPV correlation coefficients (rho) of error terms are equal to zero. In the interpretation, a positive correlation referred to complementary sources to suggest that using one source enhanced the decision to use the second source and vice versa.

## 3. Results and Discussion

### 3.1. Characteristics of Respondents

Table 2 contains the descriptive characteristics of the sampled smallholder farming households. The results are based on data obtained from 397 smallholders consisting of 197 and 200 respondents from Murang'a and Tharaka-Nithi counties, respectively. The findings revealed that the majority of respondents were male (60%), married (77%), and had at least primary-level education (94%). Therefore, married and literate male household heads dominated agricultural practice in the two regions. Agricultural production was the main occupation for most of the farming households (92%) in our study.

The average age of the sampled household heads was 52 years, with 24 years as the mean duration of farmers' experience. These results suggest that farmers accumulated knowledge of their agricultural practices and environment. Most of the sampled households had secure land tenure, as indicated by the possession of title deeds by 76% of the respondents. Farmers could, therefore, make long-term investments freely of the risk of losing ownership of their land. Only a small proportion of farmers in our sample had received formal agricultural training (25%), while the enrolment in groups and local associations was relatively low at 35% of the study sample.

**Table 2.** Descriptive statistics of respondents in the Central Highlands of Kenya.

| Variable | Mean | Standard Error |
|---|---|---|
| **Study site** | | |
| Site | 0.51 | 0.01 |
| **Farmer** | | |
| HHH gender | 0.60 | 0.03 |
| HHH literate | 0.94 | 0.01 |
| HHH married | 0.77 | 0.02 |
| HHH agriculture main occupation | 0.92 | 0.01 |
| HHH age | 52.09 | 0.77 |
| HH size | 4.08 | 0.09 |
| HHH farming experience | 24.22 | 0.77 |
| **Socio-capital** | | |
| Land secured | 0.76 | 0.02 |
| Agricultural training | 0.25 | 0.02 |
| Group membership | 0.35 | 0.02 |
| **Resources** | | |
| Arable land size | 1.32 | 0.07 |
| Tropical livestock unit (TLU) | 2.12 | 0.26 |
| Soil fertility | | |
| Soil fertility poor [a] | 0.08 | 0.01 |
| Soil fertility moderate | 0.34 | 0.02 |
| Soil fertility good | 0.58 | 0.03 |
| Soil fertility declined [b] | 0.32 | 0.02 |
| Soil fertility has no change | 0.46 | 0.03 |
| Soil fertility improved | 0.22 | 0.02 |
| Soil tested | 0.14 | 0.02 |

[a] Households with soil fertility status poor are the reference group. [b] Households with soil fertility declined are the reference.

Regarding agricultural resources, the mean size of arable land owned by farmers was 1.32 acres and the average tropical livestock units (TLU) was 2.12 units. These findings imply that farmers in the Central Highlands of Kenya had access to minimal agricultural resources, consistent with Otieno et al. [20] and Mairura et al. [60].

Concerning soil fertility status, 58% of farmers in our study held the perception that levels of fertility were good, while 34% thought that their land had moderately fertile soils. Only 8% of the respondents perceived their land as having poor soils. Whereas 46% of farmers in our study did not perceive any fluctuation in levels of soil fertility on their agricultural land, 32% and 22% of farmers thought that soil fertility was improving and declining, respectively. Only 14% of farmers in our study reported the use of soil testing services, suggesting the reliance on subjective indicators of soil conditions to inform management decisions.

*3.2. Principal Component Analysis*

The principal component analysis (PCA) revealed a Kaiser-Meyer-Olkin (KMO) measure of sampling adequacy of 0.74 and Bartlett's Test of Sphericity statistic of $p = 0.0000$ (Table 3). The KMO was greater than 0.5, and Bartlett's Test of Sphericity was statistically significant, thus signifying that the dimensional data reduction was credible [52,53]. We performed an orthogonal rotation (Varimax method) to extract the loadings matrix. We used loadings greater than 0.5 for the interpretation of results [54]. We extracted seven principal components, all of which explained a cumulative variance of 62.31% (Table 3).

**Table 3.** Extracted principal components of farmers' soil fertility information sources.

| Information Source | Information-Seeking Behaviour Principal Components (PCs) | | | | | | |
|---|---|---|---|---|---|---|---|
| | 1 | 2 | 3 | 4 | 5 | 6 | 7 |
| Family members | **0.77** | 0.10 | 0.03 | −0.09 | 0.02 | 0.10 | −0.19 |
| Friends | **0.81** | 0.09 | −0.01 | 0.09 | 0.02 | −0.02 | 0.07 |
| Neighbours | **0.83** | 0.04 | 0.01 | 0.00 | 0.15 | 0.05 | 0.17 |
| Other farmers | **0.54** | 0.11 | 0.05 | −0.01 | 0.15 | 0.01 | 0.42 |
| Progressive farmers | 0.21 | **0.68** | 0.15 | 0.13 | 0.01 | 0.08 | 0.15 |
| Agricultural extension officers | −0.04 | **0.51** | 0.14 | −0.17 | 0.21 | 0.11 | −0.05 |
| Agricultural groups | 0.08 | **0.75** | 0.02 | 0.06 | −0.06 | 0.06 | 0.16 |
| Farmers' cooperatives | 0.01 | **0.78** | −0.02 | 0.02 | 0.10 | −0.05 | 0.09 |
| Researchers | 0.05 | **0.56** | 0.05 | 0.28 | 0.34 | 0.01 | −0.24 |
| Mobile phones | 0.10 | 0.03 | **0.74** | 0.24 | 0.13 | 0.02 | −0.03 |
| Community resource centres | −0.05 | 0.17 | **0.60** | −0.19 | 0.16 | 0.38 | 0.04 |
| Internet | 0.03 | 0.19 | **0.72** | 0.31 | 0.00 | −0.05 | 0.01 |
| Agricultural shows | −0.14 | −0.09 | **0.52** | 0.18 | 0.34 | −0.10 | −0.13 |
| Newspapers | −0.05 | 0.08 | 0.17 | **0.76** | 0.08 | 0.04 | 0.07 |
| Magazines | 0.01 | 0.03 | 0.32 | **0.76** | 0.04 | 0.09 | 0.02 |
| Demonstration farms | 0.07 | 0.12 | −0.36 | **0.57** | 0.09 | 0.30 | −0.14 |
| Agro-dealers | 0.21 | 0.28 | 0.12 | 0.15 | **0.54** | −0.24 | −0.05 |
| Radio | 0.14 | 0.00 | 0.11 | 0.07 | **0.73** | 0.05 | 0.37 |
| Television | 0.07 | 0.10 | 0.08 | 0.03 | **0.73** | 0.14 | 0.01 |
| Community-based organisations | 0.08 | 0.04 | −0.02 | 0.01 | 0.06 | **0.81** | −0.03 |
| Non-governmental organisations | −0.01 | 0.05 | 0.11 | 0.37 | −0.01 | **0.58** | 0.09 |
| Faith/church-based organisations | 0.18 | −0.05 | 0.16 | 0.07 | −0.11 | **0.74** | 0.11 |
| Seminars | 0.12 | 0.02 | 0.15 | 0.14 | 0.22 | 0.02 | **0.97** |
| Chief's baraza | 0.01 | 0.26 | 0.02 | −0.01 | −0.09 | 0.26 | **0.62** |
| Farmer's knowledge and experience | 0.11 | 0.13 | −0.07 | 0.04 | 0.25 | −0.11 | **0.64** |
| Eigen value | 2.57 | 2.47 | 2.10 | 2.09 | 2.07 | 1.58 | 1.45 |
| % Explained variance | 11.19 | 10.76 | 9.11 | 9.08 | 9.00 | 6.85 | 6.31 |
| % Cumulative explained variance | 11.19 | 21.95 | 31.06 | 40.14 | 49.15 | 56.00 | 62.31 |

Extraction Method: Principal Component Analysis; Rotation Method: Varimax with Kaiser Normalisation; Kaiser-Meyer-Olkin Measure of Sampling Adequacy (0.74) and Bartlett's Test of Sphericity at $p = 0.0000$. Figures in bold are the loadings greater than 0.5.

### 3.3. Sources of Soil Fertility Information and Knowledge

The seven principal components established from farmers' information sources are shown in Tables 3 and 4. Sources derived from farmers' social networks, with high loadings on family members (0.77), friends (0.81), neighbours (0.83), and other farmers (0.54) dominated the first PC, referred to as local interpersonal sources. The information sources with high loadings under the second PC were progressive farmers (0.68), agricultural extension workers (0.51), farmers' agricultural groups (0.75), farmers' cooperatives (0.78), and researchers (0.56). This consisted of sources that were more likely to offer new knowledge and technologies to farmers through face-to-face interactions, hence, the name cosmopolite interpersonal sources.

The third PC comprised the following information sources which had high loadings: mobile phones (0.74), community resource centres (0.60), internet (0.72), and agricultural shows (0.52). We referred to this PC as aggregative/modern ICT-based sources because of the reliance on modern information and communication technologies and the fact that individual sources in this group were some kind of one-stop-shop where farmers could potentially obtain information on diverse topics. The sources falling under the fourth PC, which we referred to as print media and demonstration sources, had high loadings on newspapers (0.76), magazines (0.76), and demonstrations (0.57). The composition of the PC was perhaps an indication of the importance of demonstration activities in enhancing farmers' understanding of information obtained via printed sources.

**Table 4.** Rate of information seeking from various sources.

| PCs  Information-Seeking Behaviour | Mean | Standard Error |
|---|---|---|
| **PC 1: Local interpersonal sources** | **0.97** | **0.01** |
| Family members | 0.82 | 0.02 |
| Friends | 0.83 | 0.02 |
| Neighbours | 0.81 | 0.02 |
| Other farmers | 0.87 | 0.02 |
| **PC 2: Cosmopolite interpersonal sources** | **0.62** | **0.02** |
| Progressive farmers | 0.33 | 0.02 |
| Agricultural extension officers | 0.22 | 0.02 |
| Agricultural groups | 0.34 | 0.02 |
| Farmers Cooperatives | 0.33 | 0.02 |
| Researchers | 0.17 | 0.02 |
| **PC 3: Aggregative sources** | **0.20** | **0.02** |
| Mobile phone | 0.09 | 0.01 |
| Community resource centres | 0.03 | 0.01 |
| Internet | 0.04 | 0.01 |
| Agricultural shows | 0.14 | 0.02 |
| **PC 4: Print/demonstration sources** | **0.09** | **0.01** |
| Newspapers | 0.07 | 0.01 |
| Magazines | 0.05 | 0.01 |
| Demonstrations | 0.01 | 0.01 |
| **PC 5: Broadcast media** | **0.84** | **0.02** |
| Agro-dealers | 0.47 | 0.03 |
| Radio | 0.76 | 0.02 |
| Television | 0.44 | 0.02 |
| **PC 6: Community-based sources** | **0.27** | **0.02** |
| Community-based organisations | 0.06 | 0.01 |
| Non-governmental organisations | 0.11 | 0.02 |
| Faith-based organisations | 0.17 | 0.02 |
| **PC 7: Progressive learning** | **0.92** | **0.01** |
| Seminars | 0.01 | 0.01 |
| Chief's baraza/local public meetings | 0.39 | 0.02 |
| Farmer's knowledge and experience | 0.90 | 0.02 |

Our study established the presence of high loadings on the radio (0.73), television (0.73), and agro-dealers (0.54) under the fifth PC. The composition of this PC, known as broadcast media, suggests the important role of agro-dealers in providing inputs that were promoted via radio and television. Similarly, this combination could have been an indication that farmers turned to agro-dealers to obtain specific instructions on the use of soil fertility-enhancing inputs. The study grouped community-based organisations (0.81), non-governmental organisations (0.58), and faith-based organisations (0.74) under PC 6 identified as community-based sources. Sources in this category are usually associated with the implementation of socio-economic development projects among rural communities. Finally, the seventh PC, referred to as progressive learning sources, had high loadings on seminars (0.97), chief's baraza or local public meetings (0.62), and farmers' knowledge and experience (0.64). The sources in this group suggested self-directed learning by the farmer combined with non-formal methods of acquiring explicit knowledge.

On the usage of information sources, most of the farmers used local interpersonal sources (97%), progressive learning sources (92%), and broadcast media (84%) sources. Sixty-two per cent of the respondents mentioned the use of cosmopolite interpersonal sources. In contrast, less than 10% of the respondents reported the use of print media/demonstration sources. Similarly, there was relatively low usage of aggregative/modern ICT- and community-based sources reported by 20% and 27% of farmers, respectively.

The presence of numerous information sources accessible to farmers suggests that they had a high need for information on diverse issues consistent with Asule et al. [21]. According to Mittal and Mehar [32], farmers use different sources of information to satisfy



the same or different information needs. The complex nature of soil fertility management practices could have also increased farmers' use of information sources [17,61,62].

Several studies have also reported the significance of local interpersonal sources in providing agricultural information to smallholder farmers in developing countries [40,62,63]. Our findings corroborated those by Adolwa et al. [30] that in Ghana and Kenya, more farmers obtained information on ISFM from local networks and radio than those who used agricultural extension. Spurk et al. [9] also found similar results in a study covering the current study area and other sites in Ghana, Mali, and Zambia. Contrary to our findings, however, Gwandu et al. [36] reported that agricultural extension workers were the most important source of technical information on ISFM technologies for farmers in Zimbabwe where there were also high farmer-to-farmer interactions. The authors attributed this factor specifically to the use of participatory technology development approaches, which also enhanced farmers' interaction with other information sources such as researchers as well as their peers. In a study in India, vegetable farmers rarely used radio or television to access agricultural information, which was contrary to our findings on this aspect, although the high use of social and personal information sources was comparable to our findings [63].

Farmers turn to their peers for agricultural information because the latter are accessible and trusted sources of relevant information [64,65]. The high usage of broadcast media in our study, and especially radio, could have been due to the high accessibility of the medium. Rural farmers in Kenya have access to a range of vernacular language radio stations offering a mix of entertainment and discussion programmes centred on topical socio-economic development issues, including agriculture. Farmers will commonly tune in to their favourite programmes via their mobile phones as they carry on with their daily activities. In the study area, popular radio stations providing agricultural information were *Inooro FM* in Murang'a County and *Muuga FM* in Tharaka-Nithi County [9].

The prominence of progressive learning sources as revealed by our study suggested that farmers' knowledge and innovativeness were crucial in influencing their soil fertility management choices. Dawoe et al. [38] describe farmers' knowledge of soil fertility as consisting of a synthesis of their own experience, local knowledge systems, and information and knowledge from other farmers and extension workers [39].

In contrast to the results of our study, Cox and Sseguya [66] found that NGOs were the leading source of information on conservation agriculture for farmers in Kenya's Laikipia County, ahead of other sources like government extension agents and fellow farmers. The study also exposed the low usage of printed information sources by farmers and attributed the occurrence to low literacy levels, lack of awareness of these sources, and non-availability of relevant information. Our study's finding of low usage of community-based sources of information suggests that the numerous CBOs and NGOs in the study area were not effective in the dissemination of soil fertility information, perhaps because soil fertility improvement was not a development priority for these organisations. On modern ICTs, our findings corroborated those of other studies that rural farmers in developing countries are still lagging in the use of these technology-based sources of information despite their potential and the investment by governments to offer e-extension services [67–69].

### 3.4. Intensity of Soil Fertility Information-Seeking Behaviour

Less than 1% (2 out of 397) of respondents did not seek soil fertility information from any source, 5% sought information from sources within only one PC, and 97% used sources from more than one PC (Figure 1). Most of the smallholders, 38% (152 out of 397), sought soil fertility information from across four PCs. The findings suggest that not only did farmers require information from other sources to support their soil fertility management decisions, but they also needed to learn from diverse sources to gain the required knowledge. The implication is that a single source of information, including farmers' knowledge, was insufficient to satisfy their soil fertility information needs. Empirical evidence of the application of multiple information sources for agricultural decision-making is advanced by several other studies [30,32,70,71].

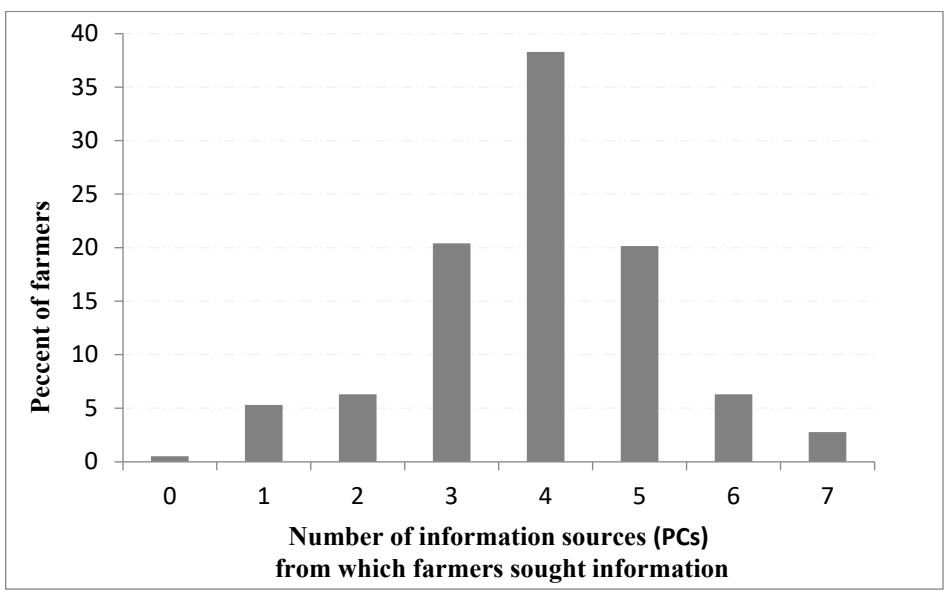

**Figure 1.** Intensity of information-seeking behaviour across the seven principal components.

*3.5. Barriers to Seeking Soil Fertility Information*

As shown in Table 5, the main obstacles faced by farmers concerning using sources of soil fertility information were the inability to identify the available sources providing the needed information (56%), insufficient information provided by sources (42%), and receiving conflicting information from the sources (38%). Other studies that identified a lack of knowledge of agricultural information sources by farmers include Brown and Llewellyn [72] and Otene et al. [73]. Farmers may fail to recognise the sources of needed information due to a lack of emphasis on soil fertility improvement as an important agricultural concern by agricultural extension agents and other information providers. On the other hand, farmers were probably complacent when it comes to seeking information because of the expectation of receiving supply-driven information services [72,74]. Related to this finding, a prominent information need established among farmers in the study area was the desire to know the sources from which to obtain soil fertility information [21]. Poor coordination of information delivery in instances where farmers have access to numerous providers could result in the dissemination of conflicting messages [29,31,75].

**Table 5.** Barriers to seeking soil fertility management information among smallholder farmers in the Central Highlands of Kenya.

| Barrier | Mean | Standard Error |
|---|---|---|
| Complex explanations | 0.14 | 0.02 |
| Conflicting information from sources | 0.38 | 0.02 |
| Unaware of information sources | 0.56 | 0.02 |
| Insufficient information | 0.42 | 0.02 |
| Language barrier | 0.08 | 0.01 |
| Farmer not interested in information seeking | 0.01 | 0.00 |
| Financial constraints | 0.01 | 0.01 |

*3.6. Covariance of Error Terms Correlation*

The likelihood ratio test (chi$^2$ = 115.332, *p* < 0.0001) of the error terms of different equations of soil fertility information sources from the HVP model was statistically significant at a 1% level of significance. Consequently, we rejected the null hypothesis that the equations were independent (Table 6). The correlation matrix showed high interdependencies among the sources of information being used by farmers, a finding that was consistent with the descriptive results where individual farmers reported the use of several sources of informa-

tion (see Figure 1). Other evidence of interdependencies among agricultural information sources used by farmers includes Mittal and Mehar [32] in India and Yaseen et al. [57] in Pakistan. Most of the variables in our analysis showed positive correlations, suggesting that farmers used their sources of soil fertility information to complement, rather than substitute, each other.

**Table 6.** Pairwise correlation coefficients across sources of soil fertility information.

| Pearson Correlations of Information Seeking Sources Combinations | Correlation Coefficient | Standard Error | Z-Value |
|---|---|---|---|
| rho21 | 0.252 * | 0.124 | 0.043 |
| rho31 | 0.112 | 0.145 | 0.440 |
| rho41 | −0.013 | 0.154 | 0.934 |
| rho51 | 0.333 * | 0.143 | 0.020 |
| rho61 | −0.066 | 0.147 | 0.654 |
| rho71 | 0.005 | 0.153 | 0.973 |
| rho32 | 0.132 | 0.107 | 0.216 |
| rho42 | 0.515 ** | 0.130 | 0.000 |
| rho52 | 0.452 ** | 0.108 | 0.000 |
| rho62 | 0.294 ** | 0.099 | 0.003 |
| rho72 | 0.625 ** | 0.135 | 0.000 |
| rho43 | 0.562 ** | 0.133 | 0.000 |
| rho53 | 0.356 * | 0.153 | 0.020 |
| rho63 | 0.128 | 0.104 | 0.217 |
| rho73 | 0.209 | 0.171 | 0.221 |
| rho54 | 0.393 ** | 0.138 | 0.004 |
| rho64 | 0.038 | 0.115 | 0.742 |
| rho74 | 0.465 ** | 0.171 | 0.006 |
| rho65 | 0.282 * | 0.110 | 0.010 |
| rho75 | 0.605 ** | 0.149 | 0.000 |
| rho76 | 0.336 * | 0.132 | 0.011 |

Likelihood ratio test of rho21 = rho31 = rho41 = rho51 = rho61 = rho71 = rho32 = rho42 = rho52 = rho62 = rho72 = rho43 = rho53 = rho63 = rho73 = rho54 = rho64 = rho74 = rho65 = rho75 = rho76 = 0: $chi^2(21) = 115.332$ Prob > $chi^2$ = 0.0000; 1 = Local interpersonal sources; 2 = Cosmopolite interpersonal sources; 3 = Aggregative sources/modern ICT-based sources; 4 = Print/demonstration sources; 5 = Broadcast media; 6 = Community-based sources; 7 = Progressive learning; ** $p \leq 1\%$; * $p \leq 5\%$.

The variable for broadcast media sources had significant positive correlations with variables for all the other information sources (PCs) considered by our study. The complementary relationships implied by this result signify the vast potential inherent in the integration of broadcast media with other sources to expand farmers' access to soil fertility information by exploiting broadcast media's wide reach and other advantages. We obtained the highest correlation coefficients for progressive learning sources and cosmopolite interpersonal sources (63%), and progressive learning and broadcast media sources (61%), making these the most compatible combinations of information sources for farmers in our study. Furthermore, cosmopolite interpersonal sources were compatible with all the other information sources except aggregative/modern ICT-based sources. In the latter case, farmers obtaining soil fertility information from cosmopolite interpersonal sources were also likely to use the other sources mentioned in our study except for modern ICT-based sources, and vice versa.

The foregoing results indicate that sources enabling access to knowledge from farmers' experience and the local conditions combined with sources providing technical knowledge were viable options for the acquisition of required soil fertility knowledge. Specifically, the results emphasise the importance of progressive learning, broadcast media, and cosmopolite interpersonal sources in line with the earlier descriptive analysis. The lack of a significant association between cosmopolite interpersonal sources and aggregative/modern ICT-based sources could be due to the poor integration of new communication technologies

within the extension system, perhaps caused by constraints on both the supply and demand sides affecting the two categories of information sources [29,66,76].

### 3.7. Determinants of Simultaneous Use of SFM Information Sources

The heptavariate probit model fits well (Wald chi$^2$ (112) = 296.72, prob > chi$^2$ = 0.0000, and log pseudo-likelihood −838.421). As a result, we rejected the null hypothesis of independence of the sources used by farmers in our study to obtain soil fertility information. Our results, instead, revealed that farmers' sources of soil fertility information in the Central Highlands of Kenya were interdependent and the use of the individual probit model for our analysis would produce biased estimates.

Our analysis revealed the factors influencing the simultaneous use of sources of soil fertility information by farmers in the Central Highlands of Kenya. The factors were location; characteristics of the household and household head (marital status, age of household head, and having agriculture as the main occupation); and socio-capital factors (access to agricultural training and group membership). Access to agricultural resources (size of arable land and livestock units owned); perceptions on soil fertility status and trends (soil fertility was good and soil fertility was improving), and soil testing (Tables 7 and 8) were also significant determinants of the simultaneous use of information sources by farmers in the study region. Table 7 shows the key drivers of the simultaneous use of information sources, emphasising the direction of influence.

**Table 7.** Estimates of the heptavariate probit model of information-seeking behaviour.

| Variable | Local Interpersonal Sources LI | Cosmopolite Interpersonal Sources CI | Aggregative Sources AG | Print/Visual Training Sources PR/V | Broadcast Media BM | Community Based Sources CB | Progressive Learning PROG |
|---|---|---|---|---|---|---|---|
| Study site | | | | | | | |
| Site | 0.827 ** | −0.220 | −0.917 *** | 0.236 | −0.541 *** | 0.450 *** | −0.969 *** |
| | (0.392) | (0.148) | (0.182) | (0.211) | (0.178) | (0.164) | (0.240) |
| Farmer and household factors | | | | | | | |
| HHH gender | 0.250 | 0.128 | −0.064 | 0.048 | 0.254 | 0.016 | 0.211 |
| | (0.338) | (0.149) | (0.183) | (0.212) | (0.176) | (0.170) | (0.223) |
| HHH literate | 0.048 | 0.162 | 0.754 | 0.121 | −0.036 | −0.421 | −0.220 |
| | (0.485) | (0.324) | (0.553) | (0.542) | (0.352) | (0.374) | (0.513) |
| HHH married | 0.607 * | 0.469 ** | 0.137 | 0.154 | −0.118 | 0.056 | 0.507 * |
| | (0.366) | (0.188) | (0.232) | (0.291) | (0.224) | (0.215) | (0.265) |
| HHH agriculture main occupation | 0.306 | −0.068 | −0.180 | −0.618 ** | −0.053 | 0.004 | 0.031 |
| | (0.731) | (0.282) | (0.300) | (0.316) | (0.424) | (0.292) | (0.474) |
| HHH age | −0.028 ** | −0.004 | 0.001 | 0.010 | −0.022 *** | −0.028 *** | −0.023 ** |
| | (0.013) | (0.007) | (0.009) | (0.010) | (0.008) | (0.009) | (0.009) |
| HH size | −0.064 | −0.007 | −0.006 | −0.110 | −0.050 | 0.037 | −0.072 |
| | (0.085) | (0.043) | (0.052) | (0.068) | (0.050) | (0.049) | (0.060) |
| HHH farming experience | 0.015 | 0.010 | −0.022 ** | −0.014 | 0.009 | 0.013 | 0.023 ** |
| | (0.013) | (0.007) | (0.009) | (0.010) | (0.008) | (0.009) | (0.009) |
| Socio-capital | | | | | | | |
| Land secured | 0.165 | −0.117 | 0.110 | 0.025 | −0.101 | 0.312 | 0.323 |
| | (0.374) | (0.175) | (0.211) | (0.256) | (0.211) | (0.210) | (0.262) |
| Agricultural training | −0.024 | 0.515 *** | 0.638 *** | −0.062 | −0.165 | 0.171 | −0.227 |
| | (0.414) | (0.185) | (0.204) | (0.245) | (0.216) | (0.183) | (0.267) |
| Group membership | 0.142 | 0.465 *** | −0.243 | −0.377 | −0.047 | −0.163 | 0.328 |
| | (0.361) | (0.156) | (0.193) | (0.240) | (0.187) | (0.172) | (0.251) |
| Access to resources | | | | | | | |
| Size of arable land | 0.017 | −0.007 | 0.065 | 0.092 | −0.108 * | −0.013 | −0.065 |
| | (0.107) | (0.052) | (0.058) | (0.060) | (0.061) | (0.059) | (0.070) |
| Tropical livestock unit (TLU) | 0.006 | 0.015 | 0.066 * | 0.036 ** | 0.018 | 0.027 * | −0.001 |
| | (0.043) | (0.018) | (0.036) | (0.016) | (0.035) | (0.016) | (0.027) |

<div style="text-align: center;">**Table 7.** *Cont.*</div>

| Variable | Local Interpersonal Sources LI | Cosmopolite Interpersonal Sources CI | Aggregative Sources AG | Print/Visual Training Sources PR/V | Broadcast Media BM | Community Based Sources CB | Progressive Learning PROG |
|---|---|---|---|---|---|---|---|
| Soil fertility perceptions | | | | | | | |
| Soil fertility good | 0.446 | 0.476 *** | 0.129 | −0.019 | 0.524 *** | 0.543 *** | 0.065 |
| | (0.324) | (0.145) | (0.182) | (0.211) | (0.173) | (0.172) | (0.217) |
| Soil fertility improved | −0.466 | 0.090 | 0.062 | 0.458 ** | −0.100 | 0.325 * | 0.357 |
| | (0.357) | (0.165) | (0.208) | (0.210) | (0.194) | (0.176) | (0.243) |
| Soil tested | 0.240 | 0.305 | 0.356 | 0.426 | 1.459 *** | 1.131 *** | 0.421 |
| | (0.688) | (0.245) | (0.247) | (0.279) | (0.500) | (0.229) | (0.368) |
| constant | 1.928 * | −0.649 | −1.211 * | −1.275 * | 2.386 *** | −0.475 | 2.313 *** |
| | (1.012) | (0.517) | (0.701) | (0.738) | (0.640) | (0.586) | (0.791) |
| Model wald chi-square (112) | 296.72 | | | | | | |
| Prob > chi-square | 0.0000 | | | | | | |
| Log pseudo-likelihood | −893.421 | | | | | | |
| Observations | 397 | | | | | | |

<div style="text-align: center;">Values in parentheses refer to robust standard errors; *** $p \leq 1\%$; ** $p \leq 5\%$; * $p \leq 10\%$.</div>

<div style="text-align: center;">**Table 8.** Summary of key predictors of simultaneous use of information sources.</div>

| Variable | Local Interpersonal Sources LI | Cosmopolite Interpersonal Sources CI | Aggregative Sources AG | Print/Visual Training Sources PR/V | Broadcast Media BM | Community Based Sources CB | Progressive Learning PROG |
|---|---|---|---|---|---|---|---|
| Study site | | | | | | | |
| Site | + | | - | | - | + | - |
| Household | | | | | | | |
| HHH gender | | | | | | | |
| HHH literate | | | | | | | |
| HHH married | + | + | | | | | + |
| HHH agriculture main occupation | | | | - | | | |
| HHH age | - | | | | - | - | - |
| HH size | | | | | | | |
| HHH farming experience | | | - | | | | + |
| Socio-capital | | | | | | | |
| Land secured | | | | | | | |
| Agricultural training | | + | + | | | | |
| Group membership | | + | | | | | |
| Resources | | | | | | | |
| Arable land | | | | | − | | |
| Tropical livestock unit (TLU) | | | + | + | | + | |
| Soil fertility | | + | | | | | |
| Soil fertility good | | | | | + | + | |
| Soil fertility improved | | | | + | | + | |
| Soil tested | | | | | + | + | |
| constant | + | | - | - | + | | + |

Relative to their counterparts from Murang'a County, smallholders residing in Tharaka-Nithi County were more likely to be found using local interpersonal and community-based sources simultaneously, but less likely to use aggregative/modern ICT-based sources, broadcast media, and progressive learning sources. Regional differences in soil fertility

information needs and supply structures could have contributed to the observed disparity in farmers' information-seeking behaviour between the two study sites. Asule et al. [21] reported different priorities for soil fertility information among farmers in the two counties. Additionally, the elevated exposure by farmers in Tharaka-Nithi County to participatory soil fertility research and development projects over a long time [18] could have influenced their information-seeking behaviour as a result of the interaction with different stakeholders. The evidence suggests that farmer participatory research approaches stimulate the acquisition and sharing of context-specific knowledge and technologies with and among farmers [77].

Married household heads, unlike their unmarried counterparts, were more likely to use local interpersonal, cosmopolite interpersonal, and progressive learning sources simultaneously to acquire soil fertility information and knowledge (Table 8). Joint decision-making was perhaps imperative in matters about soil fertility management, because of the importance of agriculture as the main source of sustaining household livelihoods. The combined knowledge resources available to both spouses in a household, coupled with their information-seeking capabilities, were therefore vital for enhanced decision-making. Other reports of better access to agricultural information resources by households having both spouses exist [63,78].

Farmers in our study were less likely to make simultaneous use of local interpersonal sources, broadcast media, community-based sources, and progressive learning sources with advancing age. Although the associations were relatively weak, the results, nevertheless, suggest a low interest in the acquisition of soil fertility knowledge and technologies by the older farmers, including even applying their ingenuity. Our findings deviated from some studies where older farmers relied on their own experience to substitute for the need to seek information from other sources [73,79]. A plausible explanation of our findings is that farmers appeared to lose interest in soil fertility improvement with age. On the other hand, there were probably stronger influences over farmers' soil fertility management decision-making, and these factors were not related to the acquisition of knowledge. It has been suggested that older farmers are more receptive to soil fertility recommendations that offer immediate benefits and are easy to implement [62,80]. Ndiritu et al. [78] also associate the lack of enthusiasm for learning by older farmers with short-range planning, risk aversion, and low energy. Our findings, therefore, highlight the need to identify the specific limitations that may be preventing older farmers from seeking soil fertility knowledge and technologies. Similarly, factors that are likely to stimulate interest and foster learning by this group, which dominates agricultural production in the study area, would be of interest. Maro et al. [81] and Martin et al. [82] reported similar age-related aversion to soil fertility knowledge and technologies among smallholders in western Kenya and Tanzania respectively. Ragasa and Mazunda et al. [83] found a non-linear relationship between age and access to agricultural extension services in Ethiopia.

Experienced farmers were more likely to use progressive learning sources, but less likely to obtain information using aggregative/modern ICT-based sources. Experienced farmers probably had more trust in their time-tested solutions, while modern ICT-based sources perhaps lacked the relevant information needed to address farmer-specific constraints [69]. Farmers' inability to access and use ICT-based sources could also explain our observation. Consistent with our findings, Mwalukasa et al. [58] established that experienced rice farmers in Tanzania hardly used mobile phones to access climate change-related information. Achora et al. [84] similarly reported that farmers in Kenya's Laikipia and Machakos counties preferred conventional sources to ICT-based sources when seeking information on conservation agriculture.

In this study, agricultural training had a significant positive influence on the simultaneous use of cosmopolite interpersonal and aggregative/modern ICT-based sources. Our study captured data on farmers' exposure to all kinds of agricultural training, irrespective of the purpose. These findings, therefore, imply that farmers' interactions with formal sources of information, through training, were effective in building their knowledge, skills,

and confidence to enable the use of cosmopolite and aggregative/ICT-based sources of information. The cosmopolite interpersonal sources category included farmer groups and extension agents as key information sources (Table 3). These results were consistent with the literature that farmers in SSA need appropriate skills to make effective use of channels such as farmer groups for presenting their needs and accessing the required information from a range of providers [8,28,85]. The need for farmer training to enhance the use of ICTs in accessing agricultural information has also been emphasised in other studies [86,87]. Due to the interdependencies of information sources used by farmers in our study, proficiency in the use of ICT and extension-related sources could have a knock-on effect on the use of other sources, thus increasing the opportunities for accessing information. In line with our findings, Yaseen et al. [57] observed a positive association between awareness of formal sources of information and the use of agricultural extension sources by farmers in Pakistan. The positive association between group membership and the use of cosmopolite interpersonal sources in our study underscored the need to promote effective farmer organisation to enhance access to agricultural information by farmers in the study area. Group membership provides an important platform through which farmers can access and understand agricultural information [49,53].

Livestock ownership was associated with farmers' propensity to use aggregative/modern ICT-based sources, print/demonstration sources, and community-based sources simultaneously. Based on our descriptive data, the sources in the three groups had relatively low usage levels by farmers, but the predisposition to use them by those with higher livestock units may be due to the ability to pay from the livestock earnings. Cost is an important barrier to accessing print- and ICT-based sources by rural farmers in developing countries. Adolwa et al. [30] similarly found the use of ICTs by some community development organisations to provide information to farmers. Our findings were also consistent with Wawire et al. [27], who found that livestock ownership influenced access to crop management information

Farmers who perceived their agricultural land as fertile were more inclined towards the simultaneous use of cosmopolite interpersonal, broadcast media, and community-based sources compared with their counterparts, who held the perception that their land was not fertile. These results corroborate Asule et al. [21] and Otieno et al. [20], that the perception of soil fertility status affects both the access and uptake of information. Farmers with perceived fertile soils may want to maximise productivity by applying even better soil and crop management practices [38], and hence, the search for relevant knowledge to support their actions. The identified sources offered useful information whose application may have led to positive soil fertility outcomes. In Malawi, Mponela et al. [33] attributed the higher adoption of ISFM technologies by farmers located in areas with numerous ISFM projects, relative to those with low project activity, to enhanced access to project-supported extension services. Wellard et al. [36] too obtained similar results with the uptake of sustainable agricultural technologies by farmers in Ghana, Uganda, and Malawi, where partnerships between national agricultural extension institutions and community development organisations promoted improved extension-farmer dialogue, farmer training, and enhanced access to required services. Likewise, Mwaniki et al. [41] found that radio effectively complemented agricultural extension in promoting climate change adaptation strategies in Kilifi County of Kenya.

Farmers who perceived a trend of improving soil fertility on their land were more likely to be those using print media/demonstration and community-based development sources simultaneously, relative to their counterparts with perceptions of stable or declining soil fertility. This was yet another interesting observation incorporating relatively under-utilised sources in the study area. The synergy arising from the integration of print/demonstration and community-based sources was effective in stimulating decisions that led to positive soil fertility outcomes. On the other hand, farmers in the study area were probably under-utilising potentially useful sources of information. Stefano et al. [43] reported a similar phenomenon in South Africa where only a small number of farmers used

printed information materials supplied by an NGO, but managed to improve their organic farming practices.

Soil testing positively and significantly influenced the simultaneous use of broadcast media and community-based sources, suggesting the propensity to use broadcast media and community-based sources by smallholders whose soil was tested. In their study of Bangladeshi farmers, Huang and Karimanzira [88] found that the main sources of information on the benefits of soil testing and fertiliser recommendations were service providers, NGOs, extension officers, and friends, in order of importance. In contrast to our findings, radio hardly played a role in raising awareness of soil testing among the farmers.

Finally, the positive and significant constant values obtained in our analysis suggest that all factors remaining constant, farmers were more inclined to seek soil fertility information from local interpersonal, broadcast media, and progressive learning sources. On the other hand, the negative and significant constant values for modern ICT-based and print media sources indicate that all factors remaining constant, farmers were less predisposed to the use of sources in the two groups. These results were consistent with the descriptive data.

## 4. Conclusions

This paper has identified the sources of soil fertility information used by farmers in the Central Highlands of Kenya as local interpersonal, cosmopolite interpersonal, aggregative/ICT-based, print media, broadcast media, community-based, and progressive learning sources. Farmers used several information sources simultaneously with our data also supporting complementary relationships, and hence, interdependencies, among the sources. We found positive correlations between broadcast media sources and all the other information sources considered in this study. The enhanced versatility of broadcast media, to be achieved through their integration with other information sources, raises their profile for dissemination of agricultural information to rural farmers. The high correlation of progressive learning sources with cosmopolite interpersonal sources offers another promising pathway for use in disseminating information to farmers. In contrast, the low correlation between cosmopolite interpersonal and ICT-based sources casts a shadow on the concerted efforts by the government and other agencies to integrate ICTS in agricultural extension, and this factor should be investigated further.

We identified the determinants of simultaneous use of soil fertility information sources: location, marital status, age, farming experience, agricultural training, group membership, livestock ownership (TLUs), perceptions of fertile soil, perceptions of improving soil fertility, and soil testing. These factors should be considered when designing extension and information dissemination programmes for different categories of farmers. There should be policies and programmes for addressing the information apathy of elderly farmers, who constitute the majority of the farming population in the study area; training farmers in the use of ICT-based sources of information; and building farmers' capacity for information consumption to enhance responsiveness to demand-led and pluralistic extension.

## 5. Study Limitations and Areas for Further Research

Information-seeking behaviour by smallholders is constrained by various factors which were not included in this study. The drawbacks to soil fertility information seeking by smallholders could be limited by resources, poverty, and contradicting information. Integrating an effective extension system could be key in enhancing the transfer of accurate and pro-farmer soil fertility information.

Further research is needed to identify the most appropriate extension approaches and pathways for enhancing access to soil fertility information. There is also a need to assess the role of agricultural projects in promoting soil fertility information through collaborative approaches.

**Author Contributions:** Conceptualisation, P.A.A. and G.N.; methodology, P.A.A., C.M., G.N. and F.K.N.; validation, C.M.; formal analysis, P.A.A., C.M. and F.K.N.; investigation, P.A.A., G.N. and F.K.N.; resources, F.K.N. and W.K.; data curation, P.A.A. and C.M.; writing—original draft preparation, P.A.A. and F.K.N.; writing—review and editing, C.M., C.S., F.K.N. and W.K.; visualisation, C.M. and F.K.N.; supervision, F.K.N., W.K. and G.N.; project administration, W.K; funding acquisition, C.S. and F.K.N. All authors have read and agreed to the published version of the manuscript.

**Funding:** This research was funded by the Swiss Agency for Development and Cooperation (SDC) and the Swiss National Science Foundation (SNSF) in the Swiss Programme for Research on Global Issues for Development (r4d programme), Grant No. 177582.

**Institutional Review Board Statement:** Not applicable.

**Data Availability Statement:** Data is contained within the article.

**Acknowledgments:** The authors acknowledge the Swiss Agency for Development and Cooperation (SDC) and the Swiss National Science Foundation (SNSF) in the Swiss Programme for Research on Global Issues for Development (r4d programme) for funding this study (Grant No. 177582) through the Organic Resource Management for Soil Fertility (ORM4Soil) project.

**Conflicts of Interest:** The authors declare no conflict of interest.

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
