# Peer review of "Determinants of Simultaneous Use of Soil Fertility Information Sources among Smallholder Farmers in the Central Highlands of Kenya"

_agriculture, doi:10.3390/agriculture13091729_

Round 1
Reviewer 1 Report
This is an interesting study. It can be seen that the author has done a lot of work, but there are still several shortcomings, the specific opinions are as follows:
(1) The introduction needs a modest rewrite. First, some studies related to this research, especially those involving core concepts, should have a systematic introduction. Second, compared with the existing studies, the marginal contribution of this study is not clear, which may be related to the poor review. Third, it seems that it is not clear what the key scientific problem to be solved in this study is, so it is suggested that the author clearly propose what the problem to be solved is.
(2) The research lacks in-depth theoretical analysis. There are many factors that affect farmers' behavior decision-making, such as what theory or basis these factors are selected by, what is the action mechanism of the core variable on the dependent variable, and what is the research hypothesis. These authors' studies are lacking, which is the biggest shortcoming of this study.
(3) There are obvious mutual causality between many independent variables and dependent variables in the research, which will lead to endogenous core variables. The authors do not seem to address this question, which may make the estimates of the study results unreliable.
Minor editing of English language required.
Author Response
Thank you for the detailed review. It has enriched our work.

Reviewer 2 Report
I provide some comments on the manuscript “Determinants of simultaneous use of soil fertility information sources among smallholder farmers in the Central Highlands of Kenya”
The authors provide important findings in the vast literature of adoption of sustainable practices. This paper is similar to relevant studies performed in African countries, including Kpadonou et al., (2017) and Mairura et al., (2022) which are cited in the paper. However, the dependent variable in this study are sources of information and knowledge on soil fertility management. This variable is relevant and few articles use it.
1 – In all tables the authors need include the standard deviation.
2 - The authors create hypotheses about the expected signs (independent variables). There needs to be a better explanation that supports this. In this case, I recommend see these two papers, Baumgart-Getz et al., (2012) and Foguesatto et al., (2020). The biggest weakness of the article is the absence of this explanation and the lack of information about why these variables were used in the questionnaire (in front of so many others).
3 – It is necessary explanation on the use of probit model. Why not logit?
4 – Why the authors used Pearson correlation rather Spearman correlation?
5 – In the conclusion, the authors need include in more details on avenues for further research. It is pivotal because this topic is widely analyzed.
Baumgart-Getz, A., Prokopy, L. S., & Floress, K. (2012). Why farmers adopt best management practice in the United States: A meta-analysis of the adoption literature. Journal of environmental management, 96(1), 17-25.
Foguesatto, C. R., Borges, J. A. R., & Machado, J. A. D. (2020). A review and some reflections on farmers' adoption of sustainable agricultural practices worldwide. Science of the total environment, 729, 138831.
Author Response
Thank you for the detailed review. It has enriched our work. We attach the responses to the raised suggestions/comments

Round 2
Reviewer 1 Report
I have no other comments, thank you.